# RAR: Region-Aware Point Cloud Registration

## ABSTRACT

This paper concerns the research problem of point cloud registration to find the rigid transformation to optimally align the source point set with the target one. Learning robust point cloud registration models with deep neural networks has emerged as a powerful paradigm, offering promising performance in predicting the global geometric transformation for a pair of point sets. Existing methods firstly leverage an encoder to regress a latent shape embedding, which is then decoded into a shape-conditioned transformation via concatenation-based conditioning. However, different regions of a 3D shape vary in their geometric structures which makes it more sense that we have a region-conditioned transformation instead of the shape-conditioned one. In this paper we present a Region-Aware point cloud Registration, denoted as RAR, to predict transformation for pairwise point sets in the self-supervised learning fashion. More specifically, we develop a novel region-aware decoder (RAD) module that is formed with an implicit neural region representation parameterized by neural networks. The implicit neural region representation is learned with a self-supervised 3D shape reconstruction loss without the need for region labels. Consequently, the region-aware decoder (RAD) module guides the training of the region-aware transformation (RAT) module and region-aware weight (RAW) module, which predict the transforms and weights for different regions respectively. The global geometric transformation from source point set to target one is then formed by the weighted fusion of region-aware transforms. Compared to the state-of-the-art approaches, our experiments show that our RAR achieves superior registration performance over various benchmark datasets (e.g. ModelNet40).

Point set registration is a challenging but meaningful task, which has wide application in many fields Bai et al. (2007); Bai & Latecki (2008); Myronenko & Song (2009); Ma et al. (2016); Wu et al. (2012); Klaus et al. (2006); Maintz & Viergever (1998); Besl & McKay (1992); Raguram et al. (2008); Yuille & Grzywacz (1988); Sonka et al. (2014). Most existing non-learning methods solve the registration problem through an iterative optimization process to search the optimal geometric transformation to minimize a pre-defined alignment loss between transformed source point set and target point set Myronenko et al. (2007); Ma et al. (2013; 2014); Ling & Jacobs (2005). The geometric transformation can be modeled by a specific type of parametric transformation (e.g. rotation, translation, thin-plate spline, and so on) Besl & McKay (1992). For example, one of the most commonly applied methods, iterative closest point (ICP) Besl & McKay (1992), estimates the rigid transformation based on a set of corresponding points. The ICP model, however, strongly depends on the initialization and has limited performance in choosing corresponding points. Moreover, iterative methods usually treat registration as an independent optimization process for each given pair of source and target point sets, which cannot transfer knowledge from registering one pair to another.

In recent years, deep-learning-based algorithms have been implemented in various industries and achieved great success, researchers are increasingly interested in bringing deep-learning-based solutions to the field of point set registration. Instead of directly optimizing the transformation matrix towards minimization of alignment loss in non-learning-based methods, learning-based methods usually leverage modern feature extraction technologies for feature learning and then regress the transformation matrix based on the mutual information and correlation defined on the extracted features of source and target shapes. The most recent model, deep closest point (DCP) Wang & Solomon (2019), leverages DGCNN Wang et al. (2019) for feature learning and a pointer network to perform soft matching. To refine the soft matching results to predict the final rigid transformation, the DCP model further proposes a singular value decomposition layer for fine-tuning. However, it is

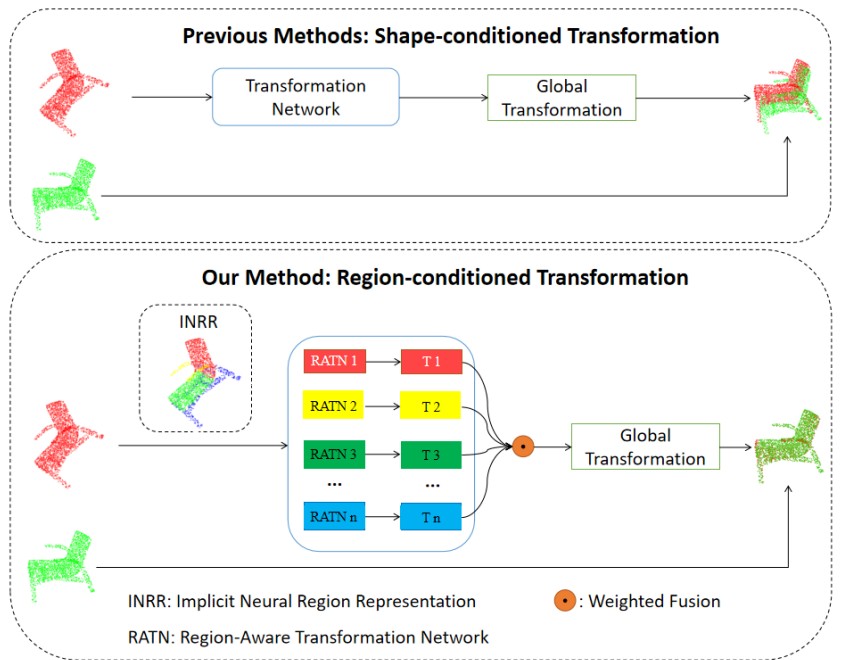

Figure 1: Comparison between the shape-conditioned transformation and region-conditioned transformation. The shape-conditioned transformation predicts one global transformation for point sets alignment whereas the region-conditioned transformation predicts a set of transformations (i.e. $\mathbf{T}n$) for different implicit regions, which are then weighted fused to form a global transformation. $\mathbf{T}n$ denotes the n-th region-specific transformation.

still challenging to design an explicit module for learning both the features from unstructured point clouds and their "geometric relationship" Wang et al. (2018). Existing works developed various models to compute the spatial correlation feature. For example, FlowNet3D Liu et al. (2019) tied to concatenate two global descriptors of source and target point sets; Balakrishnan et al. (2018) used a U-Net-based structure to mix the source and target volumetric shapes; Rocco et al. (2017) proposed a correlation tensor calculated from source and target feature map and so on. The learning of robust point cloud registration models with deep neural networks has emerged as a powerful paradigm, offering promising performance in predicting the global geometric transformation for a pair of point sets. Those methods share a similar pipeline by firstly leveraging an encoder to regress a latent shape embedding, which is then decoded into a shape-conditioned transformation via concatenation-based conditioning.

In this paper, we observe that different regions of a 3D shape vary in their geometric structures which makes it more sense that we have a region-conditioned (in contrast to shape-conditioned) transformation decoder via concatenation-based conditioning. As shown in Figure 1, the shape-conditioned transformation predicts one global transformation for point sets alignment whereas the region-conditioned transformation predicts a set of transformations for different implicit regions, which are then weighted fused to form a global transformation. With this observation, as illustrated in Figure 2, we present a region-aware point cloud registration, denoted as RAR, to predict transformation for pairwise point sets in a self-supervised learning fashion. Our proposed RAR framework contains three main components. The first component is a region-aware decoder (RAD) module that is formed with an implicit neural region representation parameterized by neural networks conditioned on a shape embedding. The implicit neural region representation is learned with a self-supervised 3D shape reconstruction loss without the need for region labels. The second component is a region-aware transformation (RAT) module which decodes shape embedding features to regress a set of region-specific transformations. The third component is the region-aware weight (RAW) module which generates the weights for different regions of the 3D shape to be aligned. The global geometric transformation from source point set to target one is then formed by weighted fusion of region-aware transforms. Our contribution is as follows:

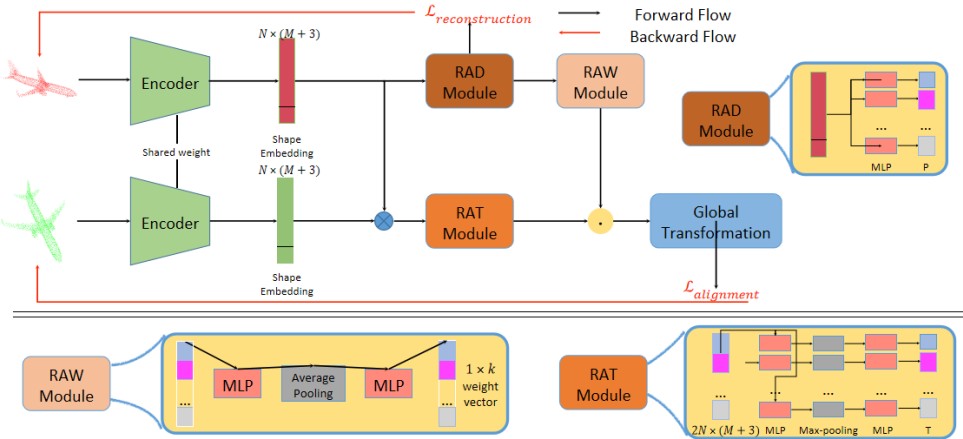

Figure 2: Our pipeline. Our proposed RAR framework contains three main components. The first component is a region-aware decoder (RAD) module that is formed with an implicit neural region representation parameterized by neural networks conditioned on a shape embedding. The implicit neural region representation is learned with a self-supervised 3D shape reconstruction loss without the need for region labels. The second component is a region-aware transformation (RAT) module which decodes shape embedding features to regress a set of region-specific transformations. The third component is the region-aware weight (RAW) module which generates the weights for different regions of the 3D shape to be aligned. The global geometric transformation from source point set to target one is then formed by weighted fusion of region-aware transforms. $\mathcal{L}_{alignment}$ denotes the alignment loss. $\mathcal{L}_{reconstruction}$ denotes the reconstruction loss which allows us to reconstruct the input shape. P denotes the region probability score and T denotes the region-specific transformation.

- We introduce a new concept of region-conditioned transformation that contributes to a novel region-aware point cloud registration (RAR) as the learning approach for robust point set alignment. Our RAR models are realized with the development of three new modules: region-aware decoder (RAD) module, region-aware transformation (RAT) module, and region-aware weight (RAW) module.

- Our RAR is a novel unsupervised learning model for point cloud registration without the need of training on labeled datasets.

- Experimental results demonstrate the effectiveness of the proposed method for point set registration, our RAR achieved superior performance compared to unsupervised and supervised state-of-the-art approaches even without labeled data for training.

# 1 RELATED WORKS

## 1.1 ITERATIVE REGISTRATION METHODS

The development of optimization algorithms to estimate rigid and non-rigid geometric transformations in an iterative routine has attracted extensive research attention in past decades. Assuming that a pair of point sets are related by a rigid transformation, the standard approach is to estimate the best translation and rotation parameters in the iterative search routine, therein aiming to minimize a distance metric between two sets of points. The iterative closest point (ICP) algorithm Besl & McKay (1992) is one successful solution for rigid registration. It initializes an estimation of a rigid function and then iteratively chooses corresponding points to refine the transformation. However, the ICP algorithm is reported to be vulnerable to the selection of corresponding points for initial transformation estimation. Go-ICP Yang et al. (2015) was further proposed by Yang et al. to leverage the BnB scheme for searching the entire 3D motion space to solve the local initialization problem brought by ICP. Zhou et al. proposed fast global registration Zhou et al. (2016) for the registration of partially overlapping 3D surfaces. The TPS-RSM algorithm was proposed by Chui and Rangarajan Chui & Rangarajan (2000) to estimate parameters of non-rigid transformations with a penalty on second-order derivatives. Existing classical algorithms have achieved great success on the registration task.

Although the independent iterative optimization process limits the efficiency of registering a large number of pairs, inspiring us to design a learning-based system for this task.

## 1.2 LEARNING-BASED REGISTRATION METHODS

In recent years, learning-based methods have achieved great success in many fields of computer vision Su et al. (2015); Sharma et al. (2016); Maturana & Scherer (2015); Bai et al. (2016); Qi et al. (2017); Verma et al. (2018); Masci et al. (2015); Zeng et al. (2017). In particular, recent works have started a trend of directly learning geometric features from cloud points (especially 3D points), which motivates us to approach the point set registration problem using deep neural networks Rocco et al. (2017); Balakrishnan et al. (2018); Zeng et al. (2017); Qi et al. (2017); Verma et al. (2018); Masci et al. (2015). PointNetLK Aoki et al. (2019) was proposed by Aoki et al. to leverage the newly proposed PointNet algorithm for directly extracting features from the point cloud with the classical Lucas & Kanade algorithm for the rigid registration of 3D point sets. Liu et al. proposed FlowNet3D Liu et al. (2019) to treat 3D point cloud registration as a motion process between points. Wang et al. proposed a deep closest point Wang & Solomon (2019) model, which first leverages the DGCNN structure to exact the features from point sets and then regress the desired transformation based on it. Balakrishnan et al. Balakrishnan et al. (2018) proposed a voxelMorph CNN architecture to learn the registration field to align two volumetric medical images. For the learning-based registration solutions listed above, the main challenge concerns how to effectively model the "geometric relationship" between source and target objects in a learning-based approach. For example, Rocco et al. (2017) proposed a correlation tensor between the feature maps of source and target images. Balakrishnan et al. (2018) leveraged a U-Net-based structure to concatenate features of source and target voxels. Liu et al. (2019); Aoki et al. (2019) used a PointNet-based structure, and Wang & Solomon (2019) used a DGCNN structure to learn the features from a point set for further registration decoding. In contrast, we introduce a region-aware point cloud registration, denoted as RAR, to predict transformation for pairwise point sets in the self-supervised learning fashion.

## 2 METHODS

We introduce our approach in the following sections. The problem statement of our method is introduced in section 2.1. We explain the learning shape descriptor in section 2.2. Section 2.3 illustrates the network structure of our region-aware decoder module. The region-aware weight module is defined in section 2.4. In section 2.5, we describe the region-aware transformation module. The loss function is also discussed in section 2.6.

## 2.1 PROBLEM STATEMENT

We define the optimization task of the deep learning-based methods which directly use unordered point clouds as input at first. Giving a training dataset $\mathbf{D} = \{(S_i, G_i)\}$, where $S_i, G_i \subset \mathbb{R}^3$. $S_i$ denotes the input source point clouds and $G_i$ denotes the input target point clouds. We aim to obtain a parametric function $g_\theta(S_i, G_i)$ using a neural network structure that can predict the rotation matrix $R \in SO(3)$ and a translation vector $t \in \mathbb{R}^3$ that can deform the source point cloud towards the target point cloud. A pre-defined alignment metric between the transformed source point cloud and the target point cloud can be defined as the objective loss function to update the parameters $\theta$. For a given dataset $\mathbf{D}$, a stochastic gradient-descent based algorithm can usually be utilized to optimize the parameters $\theta$ by minimizing the pre-defined loss function:

$$\theta^* = \arg\min_\theta [\mathbb{E}_{(S_i, G_i) \sim \mathbf{D}}[\mathcal{L}(S_i, G_i, g_\theta(S_i, G_i))]] \tag{1}$$

where $\mathcal{L}$ represents the pre-defined loss function.

## 2.2 LEARNING SHAPE EMBEDDING

For the input point clouds, the learning shape embedding is a non-linear multi-layer perceptron (MLP)-based function neural network that can extract shape features and capture the geometric information. Formally, let $P_i$ denotes the input point clouds and $f_x \subset \mathbb{R}^m$ denotes the feature of $x$, $\forall x \in P_i$, where m is the dimension of output layer. Our Learning Shape Descriptor includes

two key components: encoding network and feature information. We define the encoding network $g_1 : \mathbb{R}^3 \to \mathbb{R}^m$ which uses multi-layer perceptrons (MLP) with ReLu activation function for feature extraction:

$$f_x = g_1(x)_{x \in P_i} \tag{2}$$

The feature information is combined by extracted feature and point coordinates. Specifically, $\forall x \in P_i$, we concatenate the learned feature $f_x$ with the coordinates $x$ as the combined feature $[f_x, x] \in \mathbb{R}^{(m+3)}$. Thus, the shape descriptor of input point cloud $P_i$ is: $\{[f_x, x]\}_{x \in P_i}$.

## 2.3 REGION-AWARE TRANSFORMATION

In this section, we introduce the network structures for region-aware transformation module. In order to learn the relation information between the source point cloud $S_i$ and the target point cloud $G_i$ based on the learning shape descriptor, we define the transformation network $g_2 : \mathbb{R}^{(2m+6)} \to \mathbb{R}^c$, where c is the dimension of output layer. $g_2$ is a set of non-linear MLP-based functions. We have the predicted rigid transformation matrix $\phi_{i,k}$ as:

$$\phi_{i,k} = \text{Maxpool}\{g_{2,k}([f_x, x, f_y, y]\}_{x \in S_i, y \in G_i} \tag{3}$$

where [,] denotes the operation of concatenation, k denotes the kth transformation function in Region-Aware Transformation.

## 2.4 REGION-AWARE DECODER

We aim to use the neural networks to form the implicit neural region representation of the shape, For the learning shape embedding $[f_x, x] \in \mathbb{R}^{(m+3)}$, we use the MLP-based function $g_3 : \mathbb{R}^{(m+3)} \to \mathbb{R}^k$ to learn the implicit region field for each point. Specifically, our network outputs the region probability score that indicates the likelihood that the given point is inside the particular region:

$$l_x = Softmax(g_3([f_x, x])) \tag{4}$$

After we obtained the region probability score of points, we use the MLP-based function $g_4 : \mathbb{R}^k \to \mathbb{R}^1$ with a max pool to groups the multiple regions output to form the final implicit field $q_x$ which indicates the inside-outside status of the points and allow us to reconstruct the given shape with the reconstruction loss $\mathcal{L}_R$ we will describe in 2.6:

$$q_x = \text{Maxpool}(g_4(l_x)) \tag{5}$$

## 2.5 REGION-AWARE WEIGHT

Based on the RAD output features $[f_x, x, l_x]$ of each point, we use a MLP-based function $g_5 : \mathbb{R}^{(m+k+3)} \to \mathbb{R}^h$ with average pool layer and a MLP-based function $g_6 : \mathbb{R}^h \to \mathbb{R}^k$ to learn the region-aware weight $c_k$ for the corresponding region transformation function in region-aware transformation module:

$$c_k = g_6(\text{Avgpool}\{g_5([f_x, x, l_x])\}) \tag{6}$$

After we get the region-aware weight, we use it as the weight to balance among the multiple MLPs we obtained from the region-aware transformation module. We define the final transformation matrix $\phi$ as:

$$\phi_i = \sum_{k=1}^{K} \phi_{i,k} c_k \tag{7}$$

We further compute the transformed source point set, defined as $\hat{S}_i$.

$$\hat{S}_i = \mathbf{T}_{\phi_i}(S_i) \tag{8}$$

where $\mathbf{T}_{\phi_i}$ denotes the desired geometric transformation with parameters $\phi_i$. Based on the transformed source point set $\hat{S}_i$ and the target point set $G_i$, we further introduce the loss function.

### 2.6 LOSS FUNCTION

We define the loss function in this section. As for our unsupervised learning-based method, we do not have the transformation ground truth for supervision. Therefore, we adopt Chamfer Distance, a simple but effective distance metric defined on two non-corresponding point clouds, as our registration loss function. Given the transformed source point cloud $\hat{S}_i$ and the target point cloud $G_i$ The loss function is defined as:

$$\mathcal{L}_{CD}(\hat{S}_i, G_i) = \sum_{x \in \hat{S}_i} \min_{y \in \hat{G}_i} ||x - y||_2^2 + \sum_{y \in \hat{G}_i} \min_{x \in \hat{S}_i} ||x - y||_2^2 \tag{9}$$

As for the region-aware weight module, we leverage a mean square loss to compute the inside-outside status of the sampled points, which allows us to reconstruct the shape in the output layer. Following the sampling method in Chen & Zhang (2019), we sampled points in the 3D space surrounding the input shape and the inside-outside status of the sampled points. The unsupervised reconstruction loss is defined as:

$$\mathcal{L}_R(S_i) = (f(p) - f^*(p))^2 \tag{10}$$

where $f(p)$ is the output value of the region-aware decoder module with a maxpool operation for each point in the source point cloud, and $f^*(p)$ is the ground truth inside-outside status for a point $p$.

We define the overall loss function as:

$$\mathcal{L} = \mathcal{L}_{CD}(\hat{S}_i, G_i) + \alpha \mathcal{L}_R(S_i) \tag{11}$$

where $\alpha$ denotes the weight that balances between the registration loss and reconstruction loss.

## 3 EXPERIMENTS

In this section, we conduct experiments to demonstrate the effectiveness of our proposed method on the ModelNet40 benchmark dataset. In section 3.1 we describe the preparation of the dataset in our experiment. Section 3.2 shows our experimental settings. In section 3.3, we compare different settings and initialization of our model on the ModelNet40 dataset to demonstrate the superiority of our proposed method. We perform the study on the resistance to Data Incompleteness(D.I.) noise, Point Drifts (P.D.) noise, and Data Outliers (D.O.) noise in section 3.4. Further comparison with other state-of-the-art methods is shown in section 3.5.

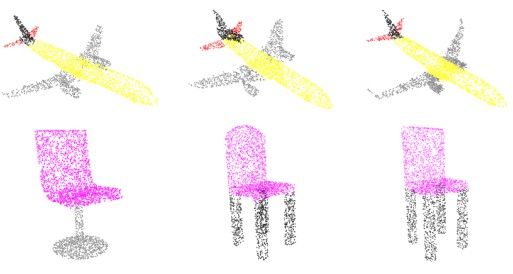

Figure 3: Qualitative results. Random selected qualitative results of region estimation.

### 3.1 DATASET PREPARATION

We test the performance of our method for 3D point cloud registration on the ModelNet40 benchmark dataset Wu et al. (2015). The ModelNet40 dataset is an open-source collection, which consists of 12311 preprocessed CAD models from 40 categories. To generate the data for training and evaluation for a fair comparison, we randomly sample 1024 points on each shape. As for each source shape, we generate the target shapes by applying a random rigid transformation: the rotation matrix is characterized by three rotation angles along the $xyz$ axes, where each value is uniformly sampled from [0, 45] unit degree, and the translation is uniformly sampled from [0.5, 0.5]. To ensure a fair comparison, we follow the official split to preprocess our dataset Wu et al. (2015).

Table 1: Quantitative result. We conduct the ablation study on the ModelNet40 dataset.

| Models | MSE(R) | RMSE(R) | MAE(R) | MSE(t) | RMSE(t) | MAE(t) |
|---|---|---|---|---|---|---|
| Baseline(one decoder) | 1.675088 | 1.289137 | 1.001619 | 0.000289 | 0.017017 | 0.012868 |
| With RAT | 1.442023 | 1.200014 | 0.961432 | 0.000202 | 0.014229 | 0.010829 |
| With RAD, RAW and RAT | **0.305423** | **0.551856** | **0.422876** | **0.000048** | **0.006938** | **0.005477** |

## 3.2 EXPERIMENTAL SETTINGS

In our proposed method, we use Adam optimizer as our optimizer. The learning shape descriptor includes 5 MLPs with dimensions of (64, 128, 128, 512, 1024). For the region-aware transformation module, we use 8 decoders for each region. Each decoder includes 3 MLPs with dimensions of (256, 128, 64), one fully connected layers with dimensions of (3) for decoding the rotation matrix, and 3 MLPs with dimensions of (256, 128, 64), one fully connected layers with dimensions of (3) for decoding the transformation matrix. The region-aware decoder module includes 3 fully connected layers with dimensions of (1024, 256, 8) to reconstruct the shape. The region-aware weight module includes 3 MLPs with dimensions of (256,256,128) and a max pool layer. Following the recent paper BAE-NET Chen et al. (2019), we set the number of pre-defined region categories as 8 and utilize one fully connected layer with dimensions of (8) to generate the final region-aware weight for region-aware transformation module. We use the ReLU activation function and implement batch normalization for every MLP layer except the last output layer. We set the loss weight $\alpha$ as 1 and the learning rate as 0.001.

For evaluation of point cloud registration performance, We use the mean squared error (MSE), root mean squared error (RMSE), and mean absolute error (MAE) to measure the performance of our model and all comparing methods. Lower values indicate better alignment performance. All angular measurements in our results are in units of degrees. Note that the ground truth labels are only used for the performance evaluation process.

## 3.3 ABLATION STUDY

**Experiment setting:** For 12311 shapes from 40 categories in the ModelNet40 dataset, We sample 1024 points from each point cloud. To show the efficiency of our proposed method, we test three settings of our model for comparison. In the first setting, without our regional-aware network, the baseline uses the classical shape-conditioned transformation network for point cloud registration. In the second setting, we leverage the RAT module which is able to predict the desired rotation and transformation matrix for multiple regions. In the third setting, we use the RAD and RAW module to predict weights for different regions respectively which is further used to weighted fuse the multiple transformations of the RAT module to get the final global transformation.

**Results:** We list the quantitative experimental results in Table 1. In this table, we examine and evaluate the performances of different models based on the prediction errors of translation vectors and rotation angles. As the process of incremental improvements that data describes, the RAD module and RAW module play significant roles in our model since they will guide the training of the RAT module. Its combination would generate outstanding performance beyond other settings, as the MSE(R) values are only 0.30. Figure 3 indicates that our method can divide the shape into different meaningful regions.

## 3.4 STUDIES ON RESISTANCE TO NOISE

**Experiment setting:** In this experiment, we conduct the experiments to verify our model's performance using Data Incompleteness (D.I.) Noise, Point Drifts (P.D.) Noise and Data Outliers (D.O.) Noise on 3D shapes. As for D.I. noise, we randomly remove a certain amount of points from the entire point set. As for P.D. noise, we randomly add Gaussian noise on the entire shapes, which is randomly sampled from N (0, 0.01) and clipped to [0.05, 0.05]. As for D.O. noise, we first remove a certain amount of points and randomly add the same amount of points generated by a zero-mean Gaussian to the entire point clouds. Note that we add the noise on the target shape. We compare our model with other representative registration methods: DCP and PR-NET. We follow exactly

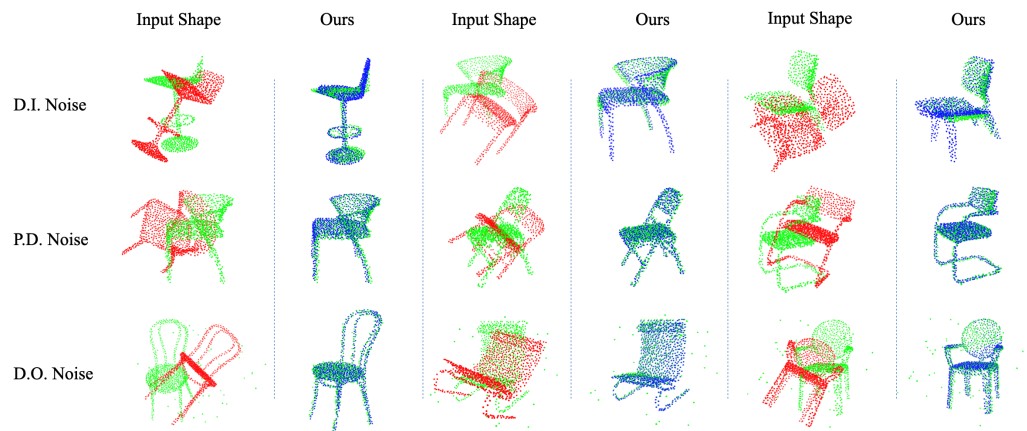

Figure 4: Qualitative results. Randomly selected qualitative results in presence of D.I., P.D., and D.O. noises on the ModelNet40 dataset.

Table 2: Quantitative result. Comparison using shapes with D.I. noise on the ModelNet40 dataset.

| Noise | Models | MSE(R) | RMSE(R) | MAE(R) | MSE(t) | RMSE(t) | MAE(t) |
|---|---|---|---|---|---|---|---|
| | PR-NET | 16.998832 | 3.930473 | 2.969286 | 0.000528 | 0.022995 | 0.017540 |
| D.I. noise | DCP | 35.178237 | 5.605362 | 4.414905 | 0.000961 | 0.031015 | 0.023442 |
| | Ours | **5.272810** | **2.286349** | **1.818420** | **0.000393** | **0.019824** | **0.014672** |
| | PR-NET | 7.700405 | 2.743103 | 2.150551 | 0.000503 | 0.022428 | 0.017721 |
| P.D. noise | DCP | 17.120563 | 3.990559 | 3.010491 | 0.000484 | 0.022016 | 0.017079 |
| | Ours | **1.734638** | **1.306637** | **1.013553** | **0.000242** | **0.015587** | **0.012142** |
| | PR-NET | 24.027700 | 4.638125 | 3.559682 | 0.000612 | 0.024748 | 0.018945 |
| D.O. noise | DCP | 53.339375 | 7.103211 | 5.779260 | 0.001221 | 0.034955 | 0.027130 |
| | Ours | **6.301775** | **2.493354** | **1.897948** | **0.000584** | **0.024166** | **0.017890** |

the source code provided by DCP, PR-NET, and their default settings for training and testing their model.

**Results:** We list the quantitative experimental results about comparison using shapes with several noises in Table 2. The table presents that our method achieves remarkably better performance than PR-NET and DCP models regarding the translation prediction and rotation angle results for the data in the ModelNet40 dataset. As for D.I. noise, our model consistently has greater accuracy far beyond others, regardless of using which error measurements. In comparison to DCP and PR-NET under P.D. noise, for all the shown metrics of alignment performance our method achieves much better results than DCP and PR-NET. Especially for the rotation matrix estimation, our method achieves an MSE of 1.67 in comparison to 15.04 achieved by DCP and 7.84 achieved by PR-NET. Note that our method is trained in a purly unsupervised manner. For the dataset in presence of D.O. noise, the quantitative result achieved by our model is better than the results of DCP and PR-NET. In comparison to the method DCP, for the precision of rotation matrix estimation, our method achieves 7.54 MSE(R) in comparison to 48.56 achieved by DCP and 23.74 achieved by PR-NET. In addition, from the qualitative results shown in Figure 4, we notice that our model achieves better alignment result with D.I. noise, P.D. noise and D.O. noise for most cases.

## 3.5 COMPARISONS WITH STATE-OF-THE-ART METHODS

**Experiment setting:** In this experiment, we evaluate the overall registration performance of our proposed model on multiple shape categories and yield the performance compared to well applied and current state-of-the-art methods. For a fair comparison, for the 12,311 CAD models from the ModelNet40, following exactly DCP's setting, we split the dataset into 9,843 models for training and 2,468 models for testing. Note that our model is trained without using any ground-truth information and our model does not require the SVD-based fine-tuning processes.

Table 3: Quantitative result. Comparison with SOTA using shapes on the ModelNet40 dataset.

| Models | MSE(R) | RMSE(R) | MAE(R) | MSE(t) | RMSE(t) | MAE(t) |
|---|---|---|---|---|---|---|
| ICP | 894.897339 | 29.914835 | 23.544817 | 0.084643 | 0.290935 | 0.248755 |
| GO-ICP | 140.477325 | 11.852313 | 2.588463 | 0.000659 | 0.025665 | 0.007092 |
| FGR | 87.661491 | 9.362772 | 1.999290 | 0.000182 | 0.013939 | 0.002839 |
| PointNetLK | 227.870331 | 15.095374 | 4.225304 | 0.000487 | 0.022065 | 0.005404 |
| DeepGMR | 7.910688 | 2.712874 | 2.812594 | 0.000100 | 0.009588 | 0.010010 |
| DCPv1+SVD | 6.480572 | 2.545697 | 1.505548 | 0.000003 | 0.001763 | 0.001451 |
| DCPv2+SVD | 1.307329 | 1.143385 | 0.770573 | 0.000003 | 0.001786 | 0.001195 |
| Ours | **0.371577** | **0.609398** | **0.475871** | 0.000070 | 0.008409 | 0.006345 |

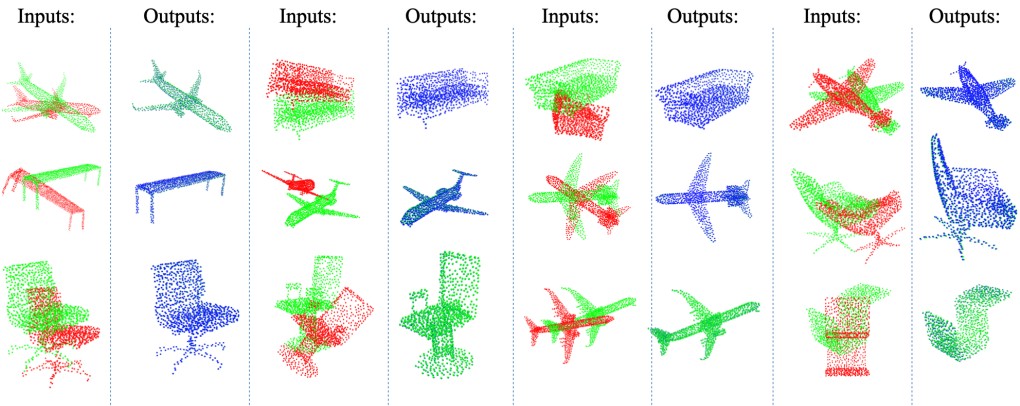

Figure 5: Qualitative results. Randomly selected qualitative results of point clouds registration on multiple categories of the ModelNet40 dataset.

**Results:** We list the quantitative experimental results in Table 3. The data demonstrate that our model, as an unsupervised method, possesses excellent generalization ability on unseen data. Even though our approach does not require label information for training purposes and an additional SVD layer for fine-tuning, our model still has significantly better performances than DCPv2+SVD (supervised) version. Also, our model is more robust to random point sampling of source and target shapes by adopting the Chamfer Distance loss, whereas the DCP would have severe degradation since it by default assumes the same sampling of points. One may note that the translation vector prediction performance of our model is inferior to that of DCPv1+SVD, DCPv2+SVD. The reason for this gap is that DCPv2+SVD adopts an additional attention mechanism in its network for enhancement. DCPv1/DCPv2+SVD leverage SVD as an additional fine-tuning the process to refine their results. Comparing to other unsupervised algorithms, like ICP and FGR, the strength and accuracy of our model could be clearly observed since ICP has an MSE(R) of 894.89 and we only have 0.37. As shown in Figure 5, the registration results indicate our model achieves remarkable performance in aligning the source and target point sets.

## 4 CONCLUSION

In this paper, we present a region-aware point cloud registration, denoted as RAR, to predict transformation for pairwise point sets in the self-supervised learning fashion. Compared with previous shape-conditioned transformation methods, with the proposed region-aware transformation network, our model can learn the desired geometric transformations for multiple regions in a particular shape which makes the model has great generalization ability and more robust to the noises. Our proposed method is trained in an unsupervised manner without any ground-truth labels. Also, we experimentally verified the effectiveness of our model and achieved superior registration results on the ModelNet40 3D point cloud registration dataset.

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
