# OpenReview forum: "RAR: Region-Aware Point Cloud Registration"
_ICLR.cc/2022/Conference — ICLR 2022 Submitted_

### Official Review · Reviewer_mwXA · 2021-11-02

**Correctness:** 3
**Technical Novelty And Significance:** 2
**Empirical Novelty And Significance:** 2
**Recommendation:** 3
**Confidence:** 2

**Main Review:**

The submission is not well prepared. For example:
(1) Missing title of section 1.
(2) The module names in Fig.1 and Fig.2 do not correspond to each other;
(3) The loss representations Fig.2 are not consistent with those in Sec. 2.6.
(4) In sec2.1, S_i and G_i represent the source and target point cloud respectively, while in Sec 2.2, there is a new symbol P_i is introduced while without any illustration of its relationship to S_i and G_i.
(5) Symble "f" in Eq.(2) represents features, while the symbol "f" in Eq.(10) represents a function. Also, the definition of f in Eq.(10) is not clear. Does it denote Eq.(4)~Eq. (5)?
(6) There should be space in "(256,256,128)" in Sec. 3.2.

Confusions regarding the method:
(1) How many subregions are selected in each point cloud, is the number pre-defined and fixed for all point clouds, or determined dynamically? If dynamically, according to what?
Though the visualizations in Fig.3 are quite beautiful, I am wondering how to determine the subregions? By the inside-outside status of the sampled points, indicated by Eq. (10)? What is the definition of "inside-outside" status, are they binary values, or continually formulated like signed distance function?
(2) Which dimension does the softmax in Eq.(4) operate on, the number of points, or the feature dimension k? Similarly, does the Maxpoo l in Eq.(5) operate on the number of points, since the input to Maxpool has a feature dimension of 1?
(3) The authors claim that their method is fully self-supervised, while Eq. (10) uses the ground truth inside-outside status.

Regarding the experiments:
(1) The content of Sec.3.2 is implementation details, while the title is "experimental settings".
(2) There are separate settings in Sec. 3.3, 3.4, and 3.5. While in the first experiments section (Sec. 3.3), there is no illustration of the training and testing set. Are the training and testing sets in the three subsections not the same?
(3) Too few comparison algorithms; missing citations of comparison algorithms; and the basic comparison algorithm DCP is out of data (published in 2019);
(4) What's the difference between DCPv1, DCPv2 and DCP? Why the comparison algorithms in Tab.2 is less than that in Tab.3?
(5) In Sec.3.5, the authors emphasize that they do not use SVD as a fine-tuning step and thus they achieve inferior results on translation estimation. What if the SVD is also applied to the proposed method?
(6) Experiments are only conducted on one dataset.

**Summary Of The Paper:**

This paper addresses the problem of point cloud registration, namely, given two point clouds with the same rigid shape, they aim to determine the relative pose (R and T) between them. For this purpose, they propose to predict several regional poses (determined by some subregions of the point cloud), weighted, and fuse them together to form the global pose estimation.

**Summary Of The Review:**

The paper is not well-organized. There are many conflict symbols and illustrations which introduce confusion.
The overall idea is just-so-so and it has less impact on the community from my point of view. The experiments settings are not clear, comparison algorithms are too old, and there is only one evaluation dataset. All those degrade the quality of this paper and I am not convinced by the results.

---

### Official Review · Reviewer_EHWV · 2021-11-02

**Correctness:** 3
**Technical Novelty And Significance:** 3
**Empirical Novelty And Significance:** 1
**Recommendation:** 3
**Confidence:** 4

**Main Review:**

The paper is understandably written and the method is presented such that it can likely be reproduced based on the paper. The discussed related work is comprehensive, however, it is notable that no papers from 2020 or 2021 are cited (this includes relevant prior art that might outperform the presented method).


Clarity of method
*****************

In Sec. 2.2, it is unclear to me if or how the proposed MLP uses the neighborhood information of points. Existing point-cloud specific networks, such as PointNet, PointNet++, DGCNN and others, that successfully tackle that problem, were not further evaluated. From the notation alone, it appears that each point is fed into the MLP on its own, without taking spacial neighborhood into account, something that would not be very effective.

In Sec. 2.3, what does the "i" index of \phi refer to in Eq. (3)?

In Sec. 2.4, what is the Maxpool operation in (5) performed over?

What is N, M in Fig. (2) in the Shape Embeddings; is M the same as m in Sec. (2.3)?

Overall it remains unclear to me how the supervision signal of Eq.(10) is built, and what the "regions" are that the method is to be aware of. Does the method use some decomposition of the ModelNet40 models? How is the decomposition into regions for Fig. (3) performed? From how I understand Sec. (2.6), the model predicts the inside-outside status of its input points, which on itself makes no sense, since all input points are exactly on the shape of the object.



Experiments
***********

The evaluation is performed on a standard dataset with a mostly usual evaluation protocoll. However, there is a significant weakness, which is that the source and target point clouds are *identical* and only rotated and translated. Usually, one would sample two different sets of N points from the original shape, then transform one of those, to simulate the fact that two scans are typically independent samplings of a common shape. This significantly weakens the conclusions drawn from this setup.

The paper cites none of the papers from 2020 and 2021. Among those are methods that significantly outperform DCP, and which might thus outperform the presented work. Just two examples:

    Choy, Christopher, Wei Dong, and Vladlen Koltun. "Deep global registration." Proceedings of the IEEE/CVF conference on computer vision and pattern recognition. 2020.

    Wu, Bingli, et al. "Feature Interactive Representation for Point Cloud Registration." Proceedings of the IEEE/CVF International Conference on Computer Vision. 2021.

Which method is "PR-NET"? Iit is only mentioned in the experiments, and there is no citiation.

Please also compare PR-NET and DCP in Table 1 (on the noise-free point clouds).

For the "D.I." noise, if points are randomly removed and yet 1024 points are sampled from the cloud, then this kind of noise would be expected to have very little to no effect. A better way to model DI noise would be to completely remove parts of the model in either source or destination (such as the backside when observed from a certain direction, thus emulating self-occlusion when scanning objects).


Minor
*****

Typos etc. (no need to respond).

- p.6, before Eq. (9), "cloud G_iThe loss"
- p.6, "translation is uniformly sampled from [0.5, 0.5]" should probably be "[-0.5, 0.5]"
- p.7: "regional-aware" -> "region-aware"


**Summary Of The Paper:**

This paper proposes a method for the registration of 3D point clouds using a trainable cascade of MLPs. The pipeline first computes per-point features, then proposes a transformation based on those features. The transformations are weighted using a predicted weight and combined into a final global transformation. The method is evaluated on ShapeNet40, showing good results w.r.t. prior art. The impact of some building blocks is validated in ablation studies.

**Summary Of The Review:**

While the method proposes an interesting idea, it is not yet ready for publication. First, the method description is unclear, especially in Sec. (2.2) and (2.6). Second, the experimental setup is not convincing, as identically sampled points are used for source and target shapes. Finally, several important related work from 2020 and 2021 is not discussed or compared against.

---

### Official Review · Reviewer_tsa1 · 2021-11-09

**Correctness:** 2
**Technical Novelty And Significance:** 2
**Empirical Novelty And Significance:** 2
**Recommendation:** 5
**Confidence:** 3

**Main Review:**

Strengths: It seems they have got a big improvement over the DeepGMR and DCP method.

Weakness: The writing is rather unclear and the paper is difficult to follow. First, how is the region or part information is predicted without any supervised signal. I don't think the network can learn this in a fully unsupervised way.  For example, how is it possible to get this segmented results in Figure 3. How could we distinguish the chair back/ chair legs, airplane body/wings without training an instance or semantic segmentation network.

Second, what is the purpose of defining the reconstruction loss and how will it help for the registration or region segmentation.

Third, what is the region aware means? segmentation or something else? Since the region aware is the key point for the paper, detailed explanation should be added.

Further, there are many more recent works on this topics and I belive it is necessary to compare with, For example,
"RPM-Net: Robust Point Matching using Learned Features"

There are many problems regarding the technical details, for example what is the number of the region, will it get learned from the network?

I think another problem is this proposed method cannot generalize to other scene-level datasets, like ICL-NUIM Dataset and the SUN3.

Finally, clearly the author has missed the section number for Introduction which makes the format looks weird.

**Summary Of The Paper:**

This paper addresses the problem of global rigid registration which is a fundemental problem in computer vision or graphics and they proposed a region-aware decoder and fuse the transformation predicted for all those regions.

**Summary Of The Review:**

The major problem is it is hard to understand the underlying principle of why this network can encode the region feature and how is the region feature gets learned without any supervision. Overall, I feel this paper is not ready for submission and is below the level of this conference. But I hope the authors could address my questions, I want to make sure this is no misunderstanding.

---

### Official Review · Reviewer_7prb · 2021-11-10

**Correctness:** 2
**Technical Novelty And Significance:** 2
**Empirical Novelty And Significance:** 3
**Recommendation:** 3
**Confidence:** 4

**Main Review:**

Overall, this paper is not in a ready shape and should be revised much before getting accepted. Many proposed modules are not technically sound and are not clear to follow. Below, I summarize the strengths and weaknesses.

Strengths:

1. The motivation to predict per-region transformations and to ensemble the transformations is interesting. Indeed, global registration is prone to local minima. Using an ensemble method might alleviate this issue and make the final prediction robust to poor feature matching.

Weaknesses:

1. The paper is not well organized and presented. A lot of technical details are hard to follow.

2. How is the region prediction module trained without supervision? It seems this paper proposes a pure self-supervised region prediction module. How does this module fit into this end-to-end pipeline? Specifically, I am extremely confused about Eq (4) & Eq (5) & Eq (6).

3. In Figure 3, the visualization of region segmentation seems very accurate. How is it possible to learn such a good region segmentation model without any supervision? Does this visualization use the ground-truth region labels?

4. In Eq (3), what's the consideration to get a feature descriptor of all points from both shapes? It seems this PointNet (or MLP) does not consider local features at all.

5. Why does the reconstruction help? For the reconstruction loss Eq (10), what does the "inside outside status" mean?

6. This paper only conducts comparisons to the original DCP and PRNet, which were presented two years ago. There are many recent papers achieving much better performance such as [a], [b], and [c].

[a] "PointNetLK Revisited"
[b] "DeepGMR: Learning latent Gaussian mixture models for registration"
[c] "RPM-Net: Robust point matching using learned features"

**Summary Of The Paper:**

This paper studies point cloud registration with deep neural networks. The key insight is learning a region-based transformation is more robust than learning a per-shape transformation even though shapes are rigid. The proposed method follows a DCP pipeline except that it performs point cloud reconstruction as well as self-supervised region segmentation. First, it uses a PointNet to lift points into a high-dimensional space and tag each point with a pseudo label. Then, points are clustered based on the region labels. Next, a per-region transformation is predicted based on the features of points in each region. Finally, the global transformation is an ensemble of region transformations.

**Summary Of The Review:**

This paper is not straight-forward to follow. Many technical details are either not clear or not sound. So it's hard to understand and justify the contributions and novelty introduced by this paper. In addition, another issue is the correctness of the region segmentation module. I don't think it can be trained without any supervision.

---

### Official Review · Reviewer_bd2E · 2021-11-10

**Correctness:** 2
**Technical Novelty And Significance:** 3
**Empirical Novelty And Significance:** 2
**Recommendation:** 3
**Confidence:** 4

**Main Review:**

This paper presents an interesting idea, i.e., adopting a region-aware estimation for the registration task. However, the authors failed to mention why such a region-aware approach can facilitate the registration task, a task that estimates transformation performed on the whole shapes. The statement "different regions of a 3D shape vary in their geometric structures which makes it more sense that we have a region-conditioned transformation instead of the shape-conditioned one" is too vague to support this motivation. Besides, this paper suffers from the following issues:

Method：
1. The proposed RAT is claimed to "predict the transforms for different regions respectively." However, as the method described in Section 2.3, it just adopts k MLP functions on the global features to obtain k transformations. Those transformations are then combined into one transformation that is performed on the whole shape instead of different regions. Thus, these k transformations are totally irrelevant with the k regions estimated in Section 2.4 and the main statement of this paper cannot be proven.
2. In Section 2.5, there are no constraints on the region-aware weights. Thus, they can be negative, which is meaningless.
3. This paper fails to mention how to obtain the translation t. Strangely, the translation errors vary in the ablation study shown in Table 1.


Experiments:
1. This method should be compared with the SOTA method RPM-Net: Robust Point Matching using Learned Features (CVPR2020).
2. The authors did not provide visual comparisons.
3. It is better to provide an ablation study for the region number k.
4. One possible advantage for this region-aware registration is to perform registration on partial shapes. However, this paper lacks such a validation.

Another major problem in this paper is the unreadable expression, including the notations, equations, descriptions, and flowchart visualization.

1. In Section 2.1, the authors defined Si and Gi as the pair of input point clouds. However, in Section 2.2, the authors suddenly used Pi, which I think the authors may refer to Si or Gi.
2. According to the flowchart in Figure 2, the RAD module is applied to Si, which failed to be implied in Section 2.4. Besides, if the region-awarer decoder is applied on Si, which is the input point cloud, then "q_x which indicates the inside-outside status" should be zero directly. If so, why do we need to calculate q_x? Moreover, the authors should state clearly the relationship between f(p) in Eqn. (10) and q_x in Eqn. (5).
3. The flowchart in Figure 2 is too vague and omits some crucial information, such as the dimensions in the RAD module. Moreover, the first MLP in the RAW module seems to be incorrect. Based on the description in Section 2.5, it is a shared MLPs that are similar to MLPs in RAD and RAT modules.

Besides, the description for this cited paper is wrong: “FlowNet3D Liu et al. (2019) tied to concatenate two global descriptors of source and target point sets.” Please check their network architecture in Figure 3 of FlowNet3D.


**Summary Of The Paper:**

This paper aims to solve the rigid registration of 3D point clouds using a deep neural network. The key difference from previous methods is that this paper proposes a region-conditioned transformation. Specifically, this method first estimates k transformation matrices and then adopts a region segmentation module to divide the shape, which is further utilized to estimate the region-aware weights to combine the k transformations

**Summary Of The Review:**

This paper presents a region-aware deep learning framework for the point cloud registration task. It can achieve good performance
and is robust to noise. However, the authors failed to state the motivation of this approach clearly and the implementation details
cannot fulfill the claimed idea. This paper also has many flaws, such as suspicious implementations, insufficient experiments and
unclear descriptions. Therefore, I tend to reject this paper.

---

### Decision · Program_Chairs · 2022-01-20

**Decision:**

Reject

**Comment:**

This paper proposes a learning-based method for shape registration that conditions on regions of the shape rather than learning from the entire point cloud in one shot.  The reviewers point out several questions about the method, thanks to expository issues as well as missing comparisons/ablation studies.  As the authors have chosen not to submit a rebuttal, I will refer them to the original reviews for details here for additional points of improvement.